# Facile Fabrication of TiO_2_ Quantum Dots-Anchored g-C_3_N_4_ Nanosheets as 0D/2D Heterojunction Nanocomposite for Accelerating Solar-Driven Photocatalysis

**DOI:** 10.3390/nano13091565

**Published:** 2023-05-06

**Authors:** Jin-Hyoek Lee, Sang-Yun Jeong, Young-Don Son, Sang-Wha Lee

**Affiliations:** 1Chemical and Biological Engineering Department, Gachon University, Seongnam-si 13120, Republic of Korea; 71in.cmsal@gmail.com (J.-H.L.); wjdtkddbs762@naver.com (S.-Y.J.); 2Department of Biomedical Engineering, College of IT Convergence, Gachon University, Seongnam-si 13120, Republic of Korea; ydson@gachon.ac.kr

**Keywords:** TiO_2_ quantum dots, g-C_3_N_4_ nanosheet, 0D/2D heterojunction, solar-driven photocatalysis

## Abstract

TiO₂ semiconductors exhibit a low catalytic activity level under visible light because of their large band gap and fast recombination of electron–hole pairs. This paper reports the simple fabrication of a 0D/2D heterojunction photocatalyst by anchoring TiO₂ quantum dots (QDs) on graphite-like C₃N₄ (g-C₃N₄) nanosheets (NSs); the photocatalyst is denoted as TiO₂ QDs@g-C₃N₄. The nanocomposite was characterized via analytical instruments, such as powder X-ray diffraction, X-ray photoelectron spectroscopy, scanning electron microscopy, transmission electron microscopy, t orange (MO) under solar light were compared. The TiO₂ QDs@g-C₃N₄ photocatalyst exhibited 95.57% MO degradation efficiency and ~3.3-fold and 5.7-fold higher activity level than those of TiO₂ QDs and g-C₃N₄ NSs, respectively. Zero-dimensional/two-dimensional heterojunction formation with a staggered electronic structure leads to the efficient separation of photogenerated charge carriers via a Z-scheme pathway, which significantly accelerates photocatalysis under solar light. This study provides a facile synthetic method for the rational design of 0D/2D heterojunction nanocomposites with enhanced solar-driven catalytic activity.

## 1. Introduction

Photocatalytic systems that utilize solar-powered oxidation–reduction chemical reactions are attractive for a wide range of applications, including water purification, air purification, and the development of self-cleaning surfaces. In particular, semiconducting nanomaterials have been widely used in photocatalytic systems that harvest abundant solar radiation, thereby allowing the mitigation of environmental pollution [1,2,3]. For the effective application of semiconductors as photocatalysts, a reduction reaction involving photoexcited free electrons must be accompanied by an oxidation reaction involving photoinduced positive holes [4]. Therefore, the valence and conduction bands of a semiconductor photocatalyst should be located at an appropriate energy level (vs. NHE) to provide an ideal bandgap for the broad absorption of solar light [3,5,6].

The photocatalytic activity of TiO_2_ semiconductors was first reported by Fujishima and Honda in 1970. Ever since then, TiO_2_ materials have been used as the gold standard in fields related to the photocatalytic degradation of pollutants [7,8,9,10]. However, TiO_2_ exhibits low visible-light photocatalytic activity levels because of the fast recombination of electron–hole pairs and the large bandgap (3.2 eV), which can only allow absorption in the ultraviolet (UV) region of the electromagnetic spectrum (≤387.5 nm) [11,12,13]. Improving the separation efficiency of photogenerated charge carriers is one of the main challenges limiting the rational design of visible-light-active TiO_2_-based semiconductor photocatalysts. Conventional TiO_2_ semiconductor photocatalysts have a typical diameter of 20–30 nm. In contrast, TiO_2_ quantum dots (QDs) have a nanoscale diameter (≤57 nm) and a high surface-area-to-volume ratio; these properties make the TiO_2_ QDs more effective at generating reactive oxygen species, owing to the increased number of active sites for photocatalytic reactions [14,15]. However, a high surface-area-to-volume ratio leads to high surface energy and causes TiO_2_ QDs to easily form aggregates, consequently worsening their photocatalytic performance.

Among the different approaches, strategies based on the use of heterojunctions with 0D/2D or 2D/0D/2D dimensions are typically considered to be efficient methods for improving the performance of photocatalysts because they allow the uniform dispersion of 0D nanoscale particles on 2D materials [16,17,18,19,20,21,22,23,24,25,26,27]. Furthermore, the close contact between components of a 0D/2D composite accelerates charge transfer through the heterojunction interface and creates fast-moving channels from the interface to photoactive surface sites [28,29,30,31,32,33]. The ability of excited free electrons to move to a lower energy level inhibits the recombination of electron–hole charge carriers, enabling more efficient charge carrier separation [28,34]. In this regard, nanosized photoactive QDs deposited on the extensive 2D materials not only provide more available active sites, but it also suppress the fast recombination of photoexcited charge carriers.

This paper reports the simple fabrication of TiO₂ QDs-anchored graphite-like C₃N₄ (g-C₃N₄) nanosheets (NSs), denoted as TiO₂ QDs@g-C₃N₄ 0D/2D heterojunction nanocomposite. In brief, TiO₂ QDs (3–5 nm) were synthesized via the hydrothermal method, and g-C₃N₄ NSs were prepared via the double calcination of urea in air at 550 °C for 2 h. Then, TiO₂ QDs were combined with thin g-C₃N₄ NSs via a sonication method, which established intimate contact between the two components of the heterojunction. The final composite was optimized by adjusting the amount of g-C₃N₄ NSs for a fixed amount of TiO₂ QDs. The optimized composite (TiO₂ QDs@g-C₃N₄) exhibited 5.7-fold and 3.3-fold higher photocatalytic activity levels than those of g-C_3_N_4_ and TiO_2_ QD alone, respectively. The uniform distribution of the TiO_2_ QDs, without severe aggregation, and the favorable electronic structure of the 0D/2D heterojunction effectively reduced the recombination rate of electron–hole pairs, significantly improving the photocatalytic performance of the heterojunction nanocomposite.

## 2. Materials and Methods

### 2.1. Chemicals

Titanium(III) chloride (TiCl_3_, 12%), urea (CH_4_N_2_O, 98%), and ethanol (EtOH, >99.9%) were purchased from Sigma-Aldrich Corporation (St. Louis, MO, USA). Deionized (DI) water (HPLC grade) was purchased from J. T. Baker Chemical Company (Phillipsburg, NJ, USA). All reagents were of analytical grade and used without further processing.

### 2.2. Synthesis of TiO_2_ QDs and g-C_3_N_4_ NSs

A total of 2.5 mL of TiCl_3_ (12%) was added to 50 mL of ethanol; the mixture was stirred for 3 h until a transparent solution was obtained. The resultant solution was decanted into an autoclave reactor and maintained at 90 °C for 3 h. Finally, the product was collected and cleaned with anhydrous ethanol via centrifugation for 20 min at a speed of 7000× *g* rpm to purify the samples, thereby forming TiO_2_ QDs. To prepare g-C_3_N_4_ NSs, urea (10 g) was finely ground using a mortar and transferred to a crucible, which was covered and wrapped in foil. The crucible was placed in a muffle furnace and was heated twice for 2 h at 550 °C in air [35,36]. The double heating process results in the production of thin g-C₃N₄ NSs.

### 2.3. Synthesis of the TiO_2_ QDs@g-C_3_N_4_ Heterojunction

A simple sonication process was conducted to form a heterojunction between TiO_2_ QDs and g-C_3_N_4_ NSs. Briefly, the as-synthesized TiO_2_ QDs (100 mg) and g-C_3_N_4_ NSs (10 mg) were dispersed in 40 mL and 20 mL of ethanol under sonication for 30 min, respectively. Two sample solutions were mixed in a beaker and sonicated for 2 h to facilitate the formation of the 0D/2D heterojunction nanocomposite. Zeta potentials of TiO_2_ and g-C_3_N_4_ samples were measured as 1.05 ± 0.19 mV and −30.57 ± 0.95 mV, respectively, indicating the interplay of electrostatic interactions between them. Then, to purify the samples, the product was collected and cleaned with anhydrous ethanol via centrifugation for 10 min at a speed of 8000× *g* rpm. The final product of TiO_2_ QDs combined with g-C_3_N_4_ NSs (so-called TiO_2_ QDs@g-C_3_N_4_) was dried overnight at 60 °C. The whole fabrication procedure is illustrated in Figure 1.

### 2.4. Photocatalytic Test

The photocatalytic properties of the as-prepared materials (TiO_2_ QDs, g-C_3_N_4_, and TiO_2_ QDs@g-C_3_N_4_) used for the decomposition of methyl orange (MO) were examined. All photocatalytic experiments were conducted at 25 °C under simulated solar light. For each of the materials, 30 mg of the photocatalyst sample was dispersed in 50 mL of an aqueous solution containing 10 ppm of MO dye. To equilibrate adsorption and desorption of MO on the catalyst, the solution was agitated in the dark for 30 min, followed by exposure to light (100 mW/cm^2^) irradiated by a solar simulator (1000 W) equipped with Xe arc lamp with an air mass global 105 filter. At a predetermined time, 1.0 mL of sample was extracted from the reaction mixture, transferred to a vacant tube, and centrifuged at a speed of 10,000× *g* rpm in order to separate the photocatalyst. Two milliliters of DI water was added to 0.5 mL of the supernatant; a ultraviolet-visible (UV-Vis) spectrophotometer was used to detect the change in absorbance at 463 nm. To confirm the major active species in the degradation of MO dye, photocatalytic reactions were conducted in the presence of three different scavengers under solar light. Isopropyl alcohol (IPA, 1 mM), benzoquinone (BQ, 1 mM), and disodium ethylenediamine tetraacetate (EDTA-2Na, 1 mM) are scavenging agents for ^•^OH, ^•^O_2_^–^ and h+ species, respectively.

## 3. Results and Discussion

### 3.1. Preparation and Characterization of Samples

Scanning electron microscopy (SEM) images of g-C_3_N_4_ and TiO_2_ QDs@g-C_3_N_4_ samples are shown in Figure 1a,b. The morphology of g-C_3_N_4_ shows a sheet-like nanostructure, with an edge thickness of ~20 nm. Compared to g-C_3_N_4_ NSs, TiO_2_ QDs@g-C_3_N_4_ showed a rougher surface morphology, retaining the sheet-like structure, but possessing a thicker edge because of the deposition of the TiO_2_ QDs. The transmission electron microscopy (TEM) images in Figure 1c,d show the distinct surface morphology of g-C_3_N_4_ NSs and TiO_2_ QDs@g-C_3_N_4_. The TEM image of g-C_3_N_4_ shows a thin sheet-like structure, confirming that it is a two-dimensional (2D) material. In the TEM image of TiO_2_ QDs@g-C_3_N_4_, TiO_2_ QDs are uniformly distributed over the g-C_3_N_4_ NS without severe aggregation; the size of the TiO_2_ QDs is in the range of 3–5 nm. In order to confirm the structure of the heterojunction nanocomposite, elemental mapping analysis of TiO_2_ QDs@g-C_3_N_4_ was performed using an energy-dispersive spectrometer (EDS). As shown in Figure 1e, the mapping images indicate the well-defined spatial distribution of Ti, O, C, and N components in the composite’s structure. In addition, the co-existence of Ti and N indicates the formation of the TiO_2_ QDs@g-C_3_N_4_ heterojunction. The EDS mapping of C components is more distinct than other components are because of the background substrate of carbon tape used for fixing the powder sample, as is usual for SEM-EDS measurements.

The crystalline phases and structures of the samples were evaluated via X-ray diffraction (XRD) conducted in the 2θ range of 10°–80°, as shown in Figure 2. XRD peaks of pure g-C_3_N_4_ appeared at 2θ = 13° (100) and 27° (002), which are consistent with the XRD peak positions of bulk g-C_3_N_4_. XRD patterns of TiO_2_ QDs and TiO_2_ QDs@g-C_3_N_4_ show the same characteristic diffraction peaks located at 2θ = 25.28°, 37.80°, 48.05°, 53.89°, and 62.68°, which correspond to the (101), (004), (200), (105), and (224) crystal phases of anatase TiO_2_, respectively [37]. However, the XRD pattern of TiO_2_ QDs@g-C_3_N_4_ showed no peaks of g-C_3_N_4_ because of the relatively small amounts of g-C_3_N_4_ relative to the contents of TiO_2_ QDs. The weight ratio of g-C_3_N_4_ (10 mg) to TiO_2_ QDs (100 mg) was approximately 9.1 wt.% in the composite.

X-ray photoelectron spectroscopy (XPS) was performed to examine the binding energy and chemical composition of TiO_2_ QDs@g-C_3_N_4_. As shown in Figure 3a, the Ti 2p core-level spectra exhibited two peaks located at 458.7 and 464.5 eV, respectively. These were assigned as 2p_3/2_ and 2p_1/2_ peaks, respectively, indicating the sole existence of Ti^4+^ because no signals were observed for Ti^2+^ and Ti^3+^ species [38,39]. In the O 1s core-level spectra shown in Figure 3b, the main peak at 530.0 eV was assigned to the Ti–O bonding of anatase TiO_2_, and the shoulder peak at 531.1 eV was attributed to the –OH group on the sample’s surface [40]. Figure 3c shows that the C 1s spectra de-convoluted into two peaks located at 284.9 and 287.5 eV, which correspond to the C–C and sp^2^-bonded N–C=N groups, respectively [41,42]. Figure 3d displays that the N 1s core-level spectra de-convoluted into three peaks at 398.0, 399.4, and 400.4 eV, which are assigned to the sp^2^-hybridized C=N–C in triazine rings, sp^2^-hybridized C–N(–C)–C or C–N(–C)-H, and C–N–H groups, respectively [43]. The XPS results confirmed the successful construction of a heterojunction nanocomposite of TiO_2_ QDs and g-C_3_N_4_ NSs components.

The XPS spectra of TiO_2_ QDs and g-C_3_N_4_ NSs were compared with those of TiO_2_ QDs@g-C_3_N_4_ to investigate the influence of the interfacial effect between them. As shown in Appendix A, the C 1s core-level spectra of the samples exhibited two distinct peaks at binding energies of 284.9 and 287.5 eV, corresponding to the C–C and sp^2^-bonded N–C=N groups, respectively [41]. The intensity of C 1s core-level spectra increased due to its binding with g-C_3_N_4_ on the surface of TiO_2_ QDs. As shown in Appendix A, the N 1s core level spectra for g-C_3_N_4_ show strong triple peaks centered at 398.2, 399.4, and 400.3 eV, corresponding to the sp^2^-hybridized C=N–C in triazine rings, sp^2^-hybridized C–N(–C)–C or C–N(–C)-H, and C–N–H groups, respectively [43]. TiO_2_ QDs@g-C_3_N_4_ also exhibited N 1s core-level spectra at the same binding energy range of g-C_3_N_4_. These results signify the interfacial interaction between TiO_2_ QDs and g-C_3_N_4_, and g-C_3_N_4_ provides a high-quality interface for TiO_2_ QDs.

Brunauer–Emmett–Teller (BET) surface area analysis was performed to compare the surface area and pore size distribution of the TiO_2_ QDs and TiO_2_ QDs@g-C_3_N_4_ samples. Figure 4a shows the Type IV isotherm curves of both samples, indicating their mesoporous structures. The BET surface areas of TiO_2_ QDs and TiO_2_ QDs@g-C_3_N_4_ samples were measured as being 367.8, and 352.8 m^2^/g, respectively. The slight decrease in the surface area in the composite could be attributed to the substitution of g-C_3_N_4_ (ca. 9.1 wt.%) for high-surface TiO_2_ QDs in the total mass. However, the g-C_3_N_4_-based heterojunction provides extensive 2D support for the dispersion of TiO_2_ QDs, which is effective to prevent the self-aggregation of TiO_2_ QDs [44]. Figure 4b shows the pore size distributions of TiO_2_ QDs (average pore diameter of 3.34 nm) and the TiO_2_ QDs@g-C_3_N_4_ heterojunction (average pore diameter of 5.2 nm), respectively. This larger pore size of the composite can be ascribed to the heterojunction of TiO_2_ QDs with g-C_3_N_4_ NSs, providing an additional interspace in the range of 20–100 nm. Moreover, TiO_2_ QDs@g-C_3_N_4_ has a larger pore volume (0.46 cm^3^/g) than that of the TiO_2_ QDs (0.32 cm^3^/g). Notably, the increases in the pore size and pore volume may induce the more uniform dispersion of TiO_2_ QDs on g-C_3_N_4_ NSs, facilitating the separation of photo-excited charged carriers and leading to a high-performance heterojunction photocatalyst.

### 3.2. Photocatalytic Activity and Mechanism

The photocatalytic activity of the samples was evaluated based on the degradation of MO dye under simulated solar light. During the photocatalytic reaction, the concentration of residual MO was calculated from the measurement of the time evolution of UV-Vis absorbance at 464 nm during the reaction time. We surveyed the mass fraction of carbon nitride in the heterojunction nanocomposite. According to Appendix A, the level of photodegradation (%) of MO dye was maximal at the optimal fraction of g-C_3_N_4_ (9.1 wt.%) in the composite, which was prepared by mixing 10 mg of g-C_3_N_4_ and 100 mg of TiO_2_ QDs under sonication for 2 h. When the mass fraction of g-C_3_N_4_ in the composite is lower than the optimal value, the photocatalytic activity of the TiO_2_ QDs@g-C_3_N_4_ is gradually decreased, while the photocatalytic activity of the TiO_2_ QDs@g-C_3_N_4_ is significantly decreased when the mass fraction of g-C_3_N_4_ is larger than the optimal value, which is probably because of the shielding effect of the excessive g-C_3_N_4_. Thus, TiO_2_ QDs@g-C_3_N_4_ (9.1 wt.%) was used for further study.

According to the XPS elemental compositions of TiO_2_ QDs@g-C_3_N_4_ (Appendix A), the weight ratio of g-C_3_N_4_ to TiO_2_ QDs is calculated as 8.47%, which is lower than 9.1% (in that all TiO_2_ are loaded with g-C_3_N_4_). Basically, it is known that g-C_3_N_4_ has a molar C/N ratio of 0.75. However, urea-prepared C_3_N_4_ has often a lower molar C/N ratio (i.e., the carbon contents are relatively larger than the N contents are) [45]. For this reason, XPS elemental analysis underestimates the proportion of g-C_3_N_4_ in the composite. The actual proportion of g-C_3_N_4_ may be larger than the calculated value of 9.1% because some TiO_2_ QDs are not fully loaded with g-C_3_N_4_ NSs. According to the XPS elemental compositions of urea-prepared C_3_N_4_ (Appendix A), the C/N ratio is calculated as being 1.2 that is larger than 0.75 for the molecular structure of g-C_3_N_4_. Synthesized g-C_3_N_4_ is considered to be polymeric C_3_N_4_ (p-C_3_N_4_) with a significant loss of nitrogen atoms. The reason may be attributed to the significant destruction of the layered structure caused by two rounds of calcination at a high temperature (550 °C) for 2 h [46].

As shown in Figure 5a, the as-prepared samples exhibited different degradation efficiencies of MO dye after 120 min. The control solution without a photocatalyst showed no change in the MO concentration, indicating the negligible photolysis of MO under simulated solar light. In contrast, the MO concentration gradually decreased, owing to the catalytic action of g-C_3_N_4_ NSs, which exhibited a degradation efficiency of 40.6% after 120 min. TiO_2_ QDs and TiO_2_ QDs@g-C_3_N_4_ exhibited 30% equilibrium adsorption of MO under dark conditions. After solar light irradiation, TiO_2_ QDs@g-C_3_N_4_ exhibited a degradation efficiency of 95.57%, which was higher than that of TiO_2_ QDs (72.76%). The color of MO solution became colorless after 120 min irradiation, as shown in the inset, signifying the almost complete mineralization of MO dye. UV-vis spectra for MO degradation are shown in Appendix A.

A linear plot of ln(*C_0_/C*) vs. reaction time is shown in Figure 5b according to the pseudo-first-order kinetics of $\ln\left(C_{O}/C\right)=kt$. The rate constants of g-C_3_N_4_ NSs, TiO_2_ QDs, and TiO_2_ QDs@g-C_3_N_4_ were fitted as k = 4.19 × 10^−3^, 7.31 × 10^−3^, and 2.38 × 10^−2^ min^−1^, respectively. The rate constant for TiO_2_ QDs@g-C_3_N_4_ was 5.7-fold and 3.3-fold larger than those of g-C_3_N_4_ and TiO_2_ QDs, respectively. The electron–hole pair separation efficiency was determined via photoluminescence (PL) analysis at an excitation wavelength of 353 nm. As shown in Figure 5c, TiO_2_ QDs@g-C_3_N_4_ exhibited the lower PL emission intensity as compared to those of the other samples (g-C_3_N_4_ and TiO_2_ QDs), indicating an enhanced separation efficiency probably due to suitable band alignment of the 0D/2D heterojunction.

Pristine P25-TiO_2_ was tested to assess the efficiency of the newly synthesized materials (TiO_2_ QDs and TiO_2_ QDs@g-C_3_N_4_). As shown in Appendix A, the photocatalytic activity of pristine P25-TiO_2_ is higher than that of the TiO_2_ QDs, but lower than that of TiO_2_ QDs@g-C_3_N_4_, i.e., the degradation efficiencies of MO dye over P25 TiO_2_, TiO_2_ QDs, and TiO_2_ QDs@g-C_3_N_4_ are 87%, 73%, and 75% after 120 min irradiation, respectively. This result indicates the superiority of the heterojunction photocatalyst over pristine P25 TiO_2_, which is the gold standard for photocatalytic reactions.

The photocatalytic mechanism of the samples was determined by measuring the decomposition rate of MO and by adding three different scavengers (1 mM). Figure 5d shows the degradation efficiencies of MO dye (10 ppm) over TiO_2_ QDs@g-C_3_N_4_ in the presence of isopropanol (IPA), disodium ethylenediamine tetraacetate (EDTA-2Na), and p-benzoquinone (BQ), which can intercept reactive species of ^•^OH, h^+^, and ^•^O_2_^−^, respectively [47]. In relation to normalized MO degradation (100%) without a scavenger, the addition of BQ decreased the degradation efficiency by 83.3%, suggesting that ^•^O_2_^−^ is the dominant active species in the decomposition of MO under solar light. The addition of IPA and EDTA-2Na also led to decreases in the degradation efficiency of 29.7% and 67.5%, respectively, indicating that the positive hole (h^+^) is a more active radical species than ^•^OH is. Figure 5d was replotted in the form of C/C_o_ versus time with the kinetic plot from the scavenger test (Appendix A). It shows the pseudo-first order kinetics that were used to produce the rate constants of k = 1.0 × 10^−3^, 1.89 × 10^−3^, and 7.43 × 10^−3^ min^−1^ for BQ, EDTA-Na, and IPA scavengers, respectively.

In summary, the TiO_2_ QDs@g-C_3_N_4_ photocatalyst is strongly influenced by the presence of BQ (^•^O_2_^−^ scavenger) and EDTA-2Na (h^+^ scavenger). The heterojunction nanocomposite offers a facilitated migration path for excited charge carriers, allowing efficient separation through the heterojunction formation with a staggered electronic structure, which leads to the more activation of ^•^O^2−^ and h^+^ radicals.

In UV-Vis diffuse reflectance spectroscopy (DRS), absorption spectra of the samples were recorded within the UV-Vis region. According to the UV-Vis DRS results shown in Figure 6a, the absorption edge of TiO_2_ QDs@g-C_3_N_4_ is shifted to a longer wavelength than that of the TiO_2_ QDs (≤~380 nm), generating more charge carriers via heterojunction formation. The products of absorption coefficient (α) and photon energy (h*ν*) are plotted as a function of photon energy to provide information about the electronic and optical properties of the samples (Figure 6b). Based on the Tauc plot, the optical bandgaps of the samples were calculated as 3.02, 3.23, and 3.19 eV for g-C_3_N_4_, TiO_2_ QDs, and TiO_2_ QDs@g-C_3_N_4_, respectively. The lower bandgap of TiO_2_ QDs@g-C_3_N_4_ suggests that it has a higher capacity for light harvesting, bestowing the beneficial effect of the 0D/2D heterojunction toward expanding the wavelength of light absorption.

The TiO_2_ QDs@ g-C_3_N_4_ heterojunction with a staggered electronic structure exhibited more photoactivity than TiO_2_ QDs and g-C_3_N_4_ did alone, signifying the interplay of facilitated transport of photo-excited charge carriers. Type-II and Z-scheme heterojunction are the main interfacial transport mechanism for the g-C_3_N_4_-based heterojunction [32,33,48]. Before establishing the photocatalytic mechanism, low-energy valence band XPS was performed to identify the valence band (VB) edge potentials of the TiO_2_ and g-C_3_N_4_ components, which were estimated as 3.24 eV and 1.76 eV, respectively (Appendix A). In addition, scavenger tests indicated that ^•^O_2_^−^ and positive hole (h^+^) were determined as the main radical species in the decomposition of MO dye.

For the Type II heterojunction, photo-induced holes in the VB position of TiO_2_ QDS are transferred to the VB position of the g-C_3_N_4_, and photo-excited electrons in the CB of g-C_3_N_4_ are transferred to the CB of TiO_2_ QDs. In the case of Type II, the valence band edge potential is not sufficient to form hydroxyl radicals from water via a reaction with positive holes because the VB position of g-C_3_N_4_ is higher than the potential of the H_2_O/•OH couple (2.8 V vs. NHE) [49]. In the Z-scheme, however, the photo-induced holes tend to stay in the more positive VB of TiO_2_ QDs, which is sufficient to produce hydroxy radicals from water, and photo-excited electrons are accumulated in the more negative CB of the g-C_3_N_4_, which maintains the high redox powers of free charge carriers. In this regard, the Z-scheme mechanism is more appropriate to interpret the photocatalytic activity of TiO_2_ QDs@g-C_3_N_4_ with highly enhanced photocatalytic activity.

In summary, the TiO_2_ QDs@ g-C_3_N_4_ heterojunction system exhibited more enhanced photoactivity than TiO_2_ QDs and g-C_3_N_4_ did alone. Photoexcited electrons in the conduction band (CB) of TiO_2_ (0.01 eV vs. NHE) are transferred to the VB of g-C_3_N_4_ (1.76 eV vs. NHE) via a Z-scheme pathway and further excited to the CB of g-C_3_N_4_ (−1.28 eV vs. NHE) with high reducing power, whereas positive holes remained in the VB of TiO_2_ QDs (3.24 eV vs. NHE), which can directly produce hydroxy radicals from water. The 0D/2D Z-scheme heterojunction causes the efficient separation of photogenerated charge carriers with high redox power, significantly enhancing solar-driven photocatalysis [8,32,33]. The photocatalytic mechanism underlying the solar-driven photocatalysis of TiO_2_ QDs@g-C_3_N_4_ is illustrated in Figure 7.

The photocatalytic stability of TiO_2_ QDs@g-C_3_N_4_ was tested by measuring the degradation efficiency of MO dye after 120 min using recycled photocatalysts. The tested sample was collected via centrifugation after the reaction. After washing it with water and ethanol several times, the recovered sample was dried in an oven for the subsequent photocatalytic reaction. The photocatalytic results for a total of four cycles are shown in Appendix A. The photocatalytic activity decreased by 2.3% even after four recycling test. TiO_2_ QDs@g-C_3_N_4_ exhibited photocatalytic stability under repeated solar light exposure, indicating the high structural stability of the heterojunction nanocomposite.

## 4. Conclusions

In this paper, we reported the facile fabrication of a TiO₂ QD-anchored g-C₃N₄ NSs, TiO₂ QDs@g-C₃N₄ 0D/2D heterojunction photocatalyst. In the TEM image of TiO_2_ QDs@g-C_3_N_4_, TiO_2_ QDs (3–5 nm) were uniformly distributed over g-C_3_N_4_ NSs, without severe aggregation. The XRD results for TiO_2_ QDs@g-C_3_N_4_ showed the same characteristic diffraction peaks at 2θ = 25.28°, 37.80°, 48.05°, 53.89°, and 62.68°, corresponding to the (101), (004), (200), (105), and (224) crystal phases of anatase TiO_2_, respectively. Furthermore, the EDS and XPS data confirmed the successful construction of the 0D/2D heterojunction nanocomposite and the coexistence of TiO_2_ and g-C_3_N_4_ components. The performance of the as-prepared samples (TiO₂ QDs, g-C₃N₄ NSs, and TiO₂ QDs@g-C₃N₄) toward MO decomposition under simulated solar light was analyzed. The TiO₂ QDs@g-C₃N₄ photocatalyst showed MO degradation, with an efficiency of 95.57%, which was 3.3-fold and 5.7-fold higher than those of TiO₂ QDs and g-C₃N₄, respectively. The 0D/2D TiO₂ QDs@g-C₃N₄ photocatalyst possessed a staggered electronic structure that facilitated the efficient separation of charge carriers via a Z-scheme pathway, significantly enhancing solar-driven photocatalysis. The work proposed a simple method for fabricating high-performance 0D/2D heterojunction photocatalysts for environmental purification and energy conversion applications.

## Data Availability

The data presented in this study are available on request from the corresponding author.

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
