# Peer review of "Facile Fabrication of TiO2 Quantum Dots-Anchored g-C3N4 Nanosheets as 0D/2D Heterojunction Nanocomposite for Accelerating Solar-Driven Photocatalysis"

_nanomaterials, 2023, doi:10.3390/nano13091565_

Round 1
Author Response
Please, see the attachment.

Reviewer 2 Report
The manuscript describes the synthesis, characterization and photocatalytic application of titanium dioxide quantum dots anchored on g-C3N4 nanosheets. The results presented show that the TiO2 QDs@g-C3N4 composite is more active in methyl orange degradation under solar irradiation than TiO2 QDs or g-C3N4 alone. The presented results are interesting, but before the manuscript can be recommended for publication, it needs to be revised.
g-C3N4 has a molar C/N ratio of 0.75. From urea prepared material has often a lower molar C/N ratio. Also, the amount of hydrogen is higher than expected for g-C3N4. The carbon nitride prepared is than called polymeric C3N4 (p-C3N4). Have the authors really synthesized graphitic carbon nitride?
The amount of titania quantum dots and that of g-C3N4 used for synthesis of the composite should be added to the “Materials and Methods” section. Why was the solution sonicated twice?
For the TiO2 QDs@g-C3N4 material, the UV/vis spectra for methyl orange degradation should be presented. Was the solution obtained after 120 min irradiation colorless? Was there also a mineralization of methyl orange observed?
From the presented data it is not clear that the reaction really followed pseudo-first order especially when using TiO2 QDs@g-C3N4. Therefore, it would be better to present the c/co versus time plots in Figure 4d instead of using the normalized degradation rate (%). What was the concentration of methyl orange and those of the scavengers in these experiments. The experiments with scavengers should be described in the “Materials and Methods” section.
The presented results should be compared with those obtained using pristine P25-TiO2 as photocatalyst to assess the efficiency of the newly synthesized material.
The valance band edge position of g-C3N4 was given with 1.76 eV. Is this potential high enough to form hydroxyl radicals from water by reaction with holes? Is there any experimental prove for the suggested mechanism?
Nothing was said about the photocatalytic stability of the TiO2 QDs@g-C3N4.
Author Response
Please, see the attachment.

Reviewer 3 Report
The manuscript describes synthesis of TiO2-g-C3N4 heterojunctions for methyl orange dye degradation. In addition to the dubious scientific novelty of the study, a large number of questions arise regarding the implementation of the experiment and the quality of the work design.
1. Element mapping should be provided in order to confirm the structure of the composite samples.
2. It is not clear why the authors took only one mass fraction of carbon nitride, it is impossible to build a study on such a small number of samples.
3. BET data should be provided.
4. Total organic carbon should be measured during the experiment.
5. TiO2 Evonik (Degussa) P25 should used as a benchmark.
6. There is very little information in the text to assert the presence of heterojunctions of a certain type. In this composite system, both heterojunctions according to type II and Z-scheme heterojunctions can arise.
In general, the article should be substantially improved, new data should be added and a full discussion should be held.
Author Response
Please, see the attachment.

Round 2
Reviewer 1 Report
The authors well revised the manuscript and it can be accepted for publication now.
Reviewer 2 Report
All my questions and comments were answered. I recommend publishing the revised manuscript.
Reviewer 3 Report
I'm satisfied with the review.